

# Reconstruction of precipitating electrons and three-dimensional structure of a pulsating auroral patch from monochromatic auroral images obtained from multiple observation points

Mizuki Fukizawa[1], Takeshi Sakanoi[1], Yoshimasa Tanaka[2, 3, 4], Yasunobu Ogawa[2, 3, 4], Keisuke Hosokawa[5], Björn Gustavsson[6], Kirsti Kauristie[7], Alexander Kozlovsky[8], Tero Raita[8], Urban Brändström[9], Tima Sergienko[9]

[1]Graduate School of Science, Tohoku University, Sendai, 980-8578, Japan
[2]National Institute of Polar Research, Tachikawa, 190-8518, Japan
[3]Polar Environment Data Science Center, Joint Support-Center for Data Science Research, Research Organization of Information and Systems, Tachikawa, 190-0014, Japan
[4]Department of Polar Science, The Graduate University for Advanced Studies (SOKENDAI), Tachikawa, 190-8518, Japan
[5]Graduate School of Informatics and Engineering, University of Electro-Communications, Chofu, 182-8585, Japan
[6]Institute for Physics and Technology, Arctic University of Norway UiT, Tromsø, 9037, Norway
[7]Finnish Meteorological Institute, Helsinki, FI-00101, Finland
[8]Sodankylä Geophysical Observatory, University of Oulu, Oulu, FI-90014, Finland
[9]IRF-Swedish Institute of Space Physics, Kiruna, SE-981 28, Sweden

*Correspondence to*: Mizuki Fukizawa (fukizawa.m@pparc.gp.tohoku.ac.jp)

**Abstract.** In recent years, aurora observation networks using high-sensitivity cameras have been developed in the polar regions. These networks allow dimmer auroras such as pulsating auroras (PsAs) to be observed with a high signal-to-noise ratio. We reconstructed the horizontal distribution of precipitating electrons using computed tomography with monochromatic PsA images obtained from three observation points. The three-dimensional distribution of the volume emission rate (VER) of the PsA was also reconstructed. The characteristic energy of the reconstructed precipitating electron flux ranged from 6 keV to 23 keV, and the peak altitude of the reconstructed VER ranged from 90 to 104 km. We evaluated the results using a model aurora and compared the model's electron density with the observed electron density. The electron density was reconstructed correctly to some extent, even after a decrease in PsA intensity. These results suggest that the horizontal distribution of precipitating electrons associated with PsAs can be effectively reconstructed from ground-based optical observations.

## 1 Introduction

Aurora computed tomography (ACT) is a method for reconstructing the three-dimensional (3-D) volume emission rate (VER) of auroral emission based on monochromatic auroral images obtained from multiple observation points (e.g., Aso et al., 1990). The horizontal distribution of precipitating electron flux can be simultaneously obtained by ACT without rocket or satellite observations (Tanaka et al., 2011). Previous studies have applied ACT to bright and well-shaped discrete auroras, such as the



quiet arc during the substorm growth phase and multiple auroral arcs (Aso et al., 1990, 1993, 1998; Frey et al., 1996; Nygrén et al., 1997; Tanaka et al., 2011). However, ACT has not been applied to pulsating auroras (PsAs).

A PsA is a type of diffuse aurora that appears as irregular patches showing quasi-periodic on–off switching of its intensity

with a periodicity of ~2–20 s (Yamamoto, 1988). The intensity is somewhat dimmer than that of a typical discrete aurora (some hundreds of R up to tens of kR at 557.7 nm; a few hundred R to ~10 kR at 427.8 nm) (McEwen et al., 1981). It has been difficult to apply ACT to PsAs because the signal-to-noise ratio (SNR) of PsA images is lower than those of discrete aurora images. However, remote operation of many high-sensitivity cameras via the internet and an archive system capable of storing a massive amount of aurora data make it possible to observe PsAs with a high SNR.

The Magnetometers Ionospheric Radars All-sky Cameras Large Experiment (MIRACLE) network consists of nine all-sky cameras (ASCs) located in the Fennoscandian region. Two of the ASCs with intensified charge-coupled devices (ICCDs) were replaced with cameras possessing the newer technology of electron-multiplying CCDs (EMCCDs) in 2007 (Sangalli et al., 2011). Ogawa et al. (2020) developed a low-cost multi-wavelength imaging system for aurora and airglow studies and installed Watec monochromatic imagers (WMIs) at several locations in the north and south polar regions. A WMI consists of a highly

sensitive CCD camera made by Watec Co., Ltd (Japan). These cameras are suitable for studying very faint auroral structures such as PsAs. In this study, we attempted to use these high-sensitivity cameras and ACT methods to reconstruct the 3-D VER of a PsA and the horizontal distribution of precipitating electrons for the first time.

## 2 Data and methods

MIRACLE ASCs observed PsA patches from Kilpisjärvi (KIL, 69.05°N, 20.36°E), Abisko (ABK, 68.36°N, 18.82°E), and

WMI ASCs at Skibotn (SKB, 69.35°N, 18.82°E) during the substorm recovery phase from 0:00 UT to 2:00 UT on 18 February 2018. These ASCs have a typical field-of-view, as shown in Fig. 1b. The position of Tromsø (TRO, 69.58°N, 19.23°E) where the European incoherent scatter (EISCAT) radar operates is also shown. We selected 427.8 nm auroral images in which PsA patches were detected at the EISCAT radar observation point. The reconstructed results were compared with the electron density observed by the EISCAT radar in Sect. 3.4. Figure 1a shows 427.8 nm auroral images obtained by the three ASCs

from 00:53:30 UT to 00:53:42 UT. The temporal resolution of each ASCs was 2 s. A median filter of 3 × 3 pixels was applied to auroral images to improve the SNR. We also composited auroral images obtained from four WMI CCD cameras of the same type at SKB. The auroral image at SKB has a time ambiguity of ~1–2 s. We determined the time when the auroral image was obtained by aligning the temporal changes in the PsA patch as shown in auroral images from ABK, KIL, and SKB.

The ACT method used in this study is based on the method proposed by Tanaka et al. (2011). We adopted an oblique coordinate

system with the origin (O) at coordinates of (69.4°N, 19.2°E). The X-axis was anti-parallel to the horizontal component of the geomagnetic field, the Y-axis was eastward, and the Z-axis was anti-parallel to the geomagnetic field and perpendicular to the



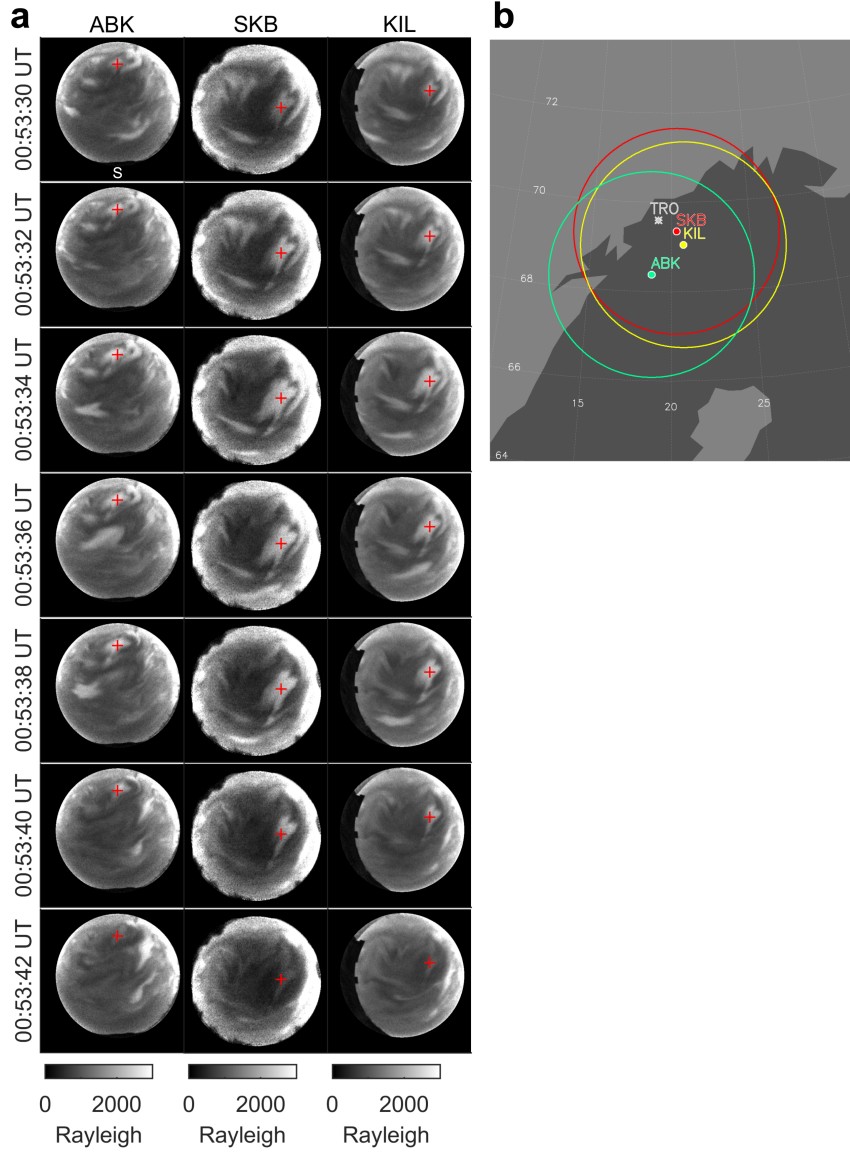

**Figure 1: (a) Successive auroral images from Abisko (ABK), Skibotn (SKB), and Kilpisjärvi (KIL) from 00:53:30 UT to 00:53:42 UT on 18 February 2018. (b) Locations and field of views of all-sky cameras at ABK (green), SKB (red), and KIL (yellow) at an altitude of 100 km. The location of Tromsø (TRO) is shown by a white asterisk.**

Y-axis (see Fig. 2 in Tanaka et al. (2011)). The simulation region ranged from −75 to 75 km, from −100 to 100 km, and from 80 to 180 km for the X-, Y-, and Z-axes, respectively. We set the energy ($E$) range to extend from 300 eV to 100 eV. This region was divided linearly into $n_x \times n_y \times n_z$ voxels along the X-, Y-, and Z-axes and logarithmically into $n_E$ bins in the $E$ direction. We set the parameters ($n_x$, $n_y$, $n_z$, $n_E$) to (75, 100, 50, 50), corresponding to a spatial mesh size of $2 \times 2 \times 2$ km. These parameters were selected so that each voxel has at least one line-of-sight crossing from the pixels in the auroral images.



The differential flux of precipitating electrons was reconstructed by maximizing the posterior probability $P(\mathbf{f}|\tilde{\mathbf{g}})$, where $\mathbf{f}$ is a vector of the differential flux of precipitating electrons and $\tilde{\mathbf{g}}$ is a vector of gray levels at pixels in the auroral images obtained with ASCs. According to Bayes' theorem, the posterior probability $P(\mathbf{f}|\tilde{\mathbf{g}})$ is given by (Tanaka et al., 2011)

$$P(\mathbf{f}|\tilde{\mathbf{g}}) \propto \exp\left[-\frac{1}{2}\left\{\left(\tilde{\mathbf{g}} - \mathbf{g}(\mathbf{f})\right)^T \Sigma^{-1}\left(\tilde{\mathbf{g}} - \mathbf{g}(\mathbf{f})\right) + \frac{\left\|\nabla^2 \mathbf{f}\right\|^2}{\sigma^2}\right\}\right],$$
(1)

where $\mathbf{g}(\mathbf{f})$ is a vector of grey levels obtained by line-integrating the VER in the line-of-sight direction from each pixel (Eq. (8) in Tanaka et al. (2011)). The VER was derived from model $\mathbf{f}$ using the aurora emission model (Eq. (3) in Tanaka et al. (2011)). $\Sigma^{-1}$ is the inverse covariance matrix, $\sigma$ is the variance of $\nabla^2 \mathbf{f}$, and the second-order derivative of $\mathbf{f}$ is taken with respect to $x, y$, and $E$. The second term in Eq. (1) represents the smoothness of $\mathbf{f}$ in space and energy directions. We determined
$\Sigma^{-1}$ as the standard deviation calculated from each auroral image. The $32 \times 32$ pixel region in which no PsA patch was contained was used to calculate $\Sigma^{-1}$. To maximize the posterior probability, it is necessary to minimize the function

$$\varphi\left(\mathbf{f}; \lambda, \lambda_E, c_j\right) = \sum_j \left(c_j \tilde{\mathbf{g}}_j - \mathbf{g}_j(\mathbf{f})\right)^T \Sigma_j^{-1} \left(c_j \tilde{\mathbf{g}}_j - \mathbf{g}_j(\mathbf{f})\right) + \lambda^2 \left\|\nabla_{x,y}^2 \mathbf{f} + \lambda_E^2 \nabla_E^2 \mathbf{f}\right\|^2,$$
(2)

where $\lambda, \lambda_E$, and $c_j$ are the so-called hyperparameters, which are constants corresponding to the weighting factors for the spatial ($\lambda$) and energy ($\lambda_E$) derivative terms and the correction factors for the relative sensitivity between cameras ($c_j$), respectively.
The subscript $j$ signifies the three observation points (ABK and KIL). The parameter $c_{SKB}$ was fixed at 1. The summation was conducted for the first term in Eq. (2) since $c_j$ and $\Sigma_j^{-1}$ were different during the three ASCs.

We carried out the change of variables $\mathbf{f} = \exp(\mathbf{x})$ to take advantage of the non-negative constraint on the differential flux $\mathbf{f}$ (i.e., $\mathbf{f} \geq 0$). We then minimized the function $\varphi(\mathbf{x}; \lambda, \lambda_E, c_{ABK}, c_{KIL})$ by implementing the Gauss–Newton algorithm with the initial value $\mathbf{x}^{(0)} = \log\left(\mathbf{f}^{(0)}\right)$, where $\mathbf{f}^{(0)} = 10^7$ [m$^{-2}$ s$^{-1}$ eV$^{-1}$].

The hyperparameters were determined using the fivefold cross-validation method (Stone, 1974). First, elements of the vector $\tilde{\mathbf{g}}$ were divided into 5 subsets. Then, one was selected as a test set ($\tilde{\mathbf{g}}_j^{tes}$) and the others as a training set ($\tilde{\mathbf{g}}_j^{tra}$). We found the solution $\hat{\mathbf{x}}$ to minimize $\varphi(\mathbf{f}; \lambda, \lambda_E, c_{ABK}, c_{KIL})$ using only the training set $\tilde{\mathbf{g}}_j^{tra}$ and then predicted the test set $\mathbf{g}_j^{tes}(\hat{\mathbf{x}})$. We then calculated the sum of the squares of the residuals between the test data and the predicted data:

$$\delta\left(w, \lambda, \lambda_E, c_j\right) = \sum_j \left\|c_j \tilde{\mathbf{g}}_j^{tes} - \mathbf{g}_j^{tes}(\hat{\mathbf{x}})\right\|^2.$$
(3)

The cross-validation score $\bar{\delta}(\lambda, \lambda_E, c_{ABK}, c_{KIL})$ was calculated by averaging over 5 $\delta(\lambda, \lambda_E, c_{ABK}, c_{KIL})$s, which were obtained by replacing the test set with one of the training sets in turn.



The hyperparameters $\lambda$, $\lambda_E$, $c_{\mathrm{ABK}}$, and $c_{\mathrm{KIL}}$ were determined by minimizing $\bar{\delta}(\lambda, \lambda_E, c_{\mathrm{ABK}}, c_{\mathrm{KIL}})$ with a trial-and-error method. In addition, the number of iterations for the Gauss-Newton algorithm was also simultaneously determined to be 200 to minimize $\bar{\delta}$.

The PsA patches shown in Fig. 1a are embedded in the background diffuse auroral emission. We found that a horizontally uniform diffuse aurora causes ambiguity in the reconstruction result because the altitude of the uniform auroral structure cannot be determined from the single-wavelength images. Thus, we subtracted the background emission from the images prior to ACT reconstruction. We created the background emission image by assuming the same value for all voxels. The background VER was taken to be 75 cm$^{-3}$ s$^{-1}$, corresponding to the spatially averaged observed background emission intensity.

## 3 Results and discussion

### 3.1 Reconstruction of a pulsating aurora patch model

We reconstructed a model PsA patch from pseudo auroral images to evaluate the analytical error of ACT before reconstructing the PsA patch from the observed auroral images. To create the pseudo auroral images, we prepared the horizontal distributions of the total energy, $Q_0$, and the characteristic energy, $E_c$. We then derived the 3-D VER, $L$, as shown in Fig. 2a. The total

energy was assumed to have a Gaussian shape in horizontal directions with a maximum value of 6 mW m$^{-2}$. The energy distribution was considered to be a Maxwellian distribution with an uniform characteristic energy of 15 keV. Pseudo auroral images were obtained from $L$ by solving the forward problem (Fig. 2b). We added random noise from a normal distribution with a mean value of 0 and the standard deviation determined from observed auroral images.

Figure 2c shows $Q_0$, $E_c$, and $L$ reconstructed from the pseudo auroral images. The values of $Q_0$ were calculated as $Q_0 =$

$\sum_i E_i f(E_i)(E_{i+1} - E_i)$. When we assume the energy distribution to be a Maxwellian distribution, the characteristic energy can be written as $E_c = \frac{1}{2}\langle E \rangle = \frac{1}{2}\frac{Q_0}{\sum_i f(E_i)(E_{i+1}-E_i)}$. We calculated the errors between the model and the result for $Q_0$, $E_c$, and $L$ (Fig. 2d). The median values of the errors were $-5\%$ for $Q_0$, $-21\%$ for $E_c$, and $-11\%$ for $L$. The northwestern part of $Q_0$ was overestimated by at most 23%, the edge part was underestimated by at most 29%, and the central part was underestimated by $\sim$8%. The central part of $E_c$ was reconstructed with similar accuracy. In comparison, the edge part (especially the northwestern

part) was underestimated by at most 56%. The underestimation of $E_c$ was caused by the overestimation of the emission altitude (Fig. 2d). Information regarding the PsA emission altitude is easily lost in obtaining the auroral image, since the structure of the PsA patch is vertically thin and horizontally wide. In addition, the SNR at the edge part is lower than at the central part since we assumed a Gaussian shape for the horizontal distribution of $Q_0$. These factors would tend to reduce the accuracy at the edge part.

It should be noted that the reconstructed results using the hyperparameters determined by the cross-validation method revealed unexpected fine structures. To avoid this phenomenon, we set the lower limit of $\lambda$ by a different method, namely by minimizing



the residual sum of squares between the model and the reconstructed result of $Q_0$ and $E_c$. The lower limit on $\lambda$ makes it challenging to reconstruct actual fine-scale structures in the patches.

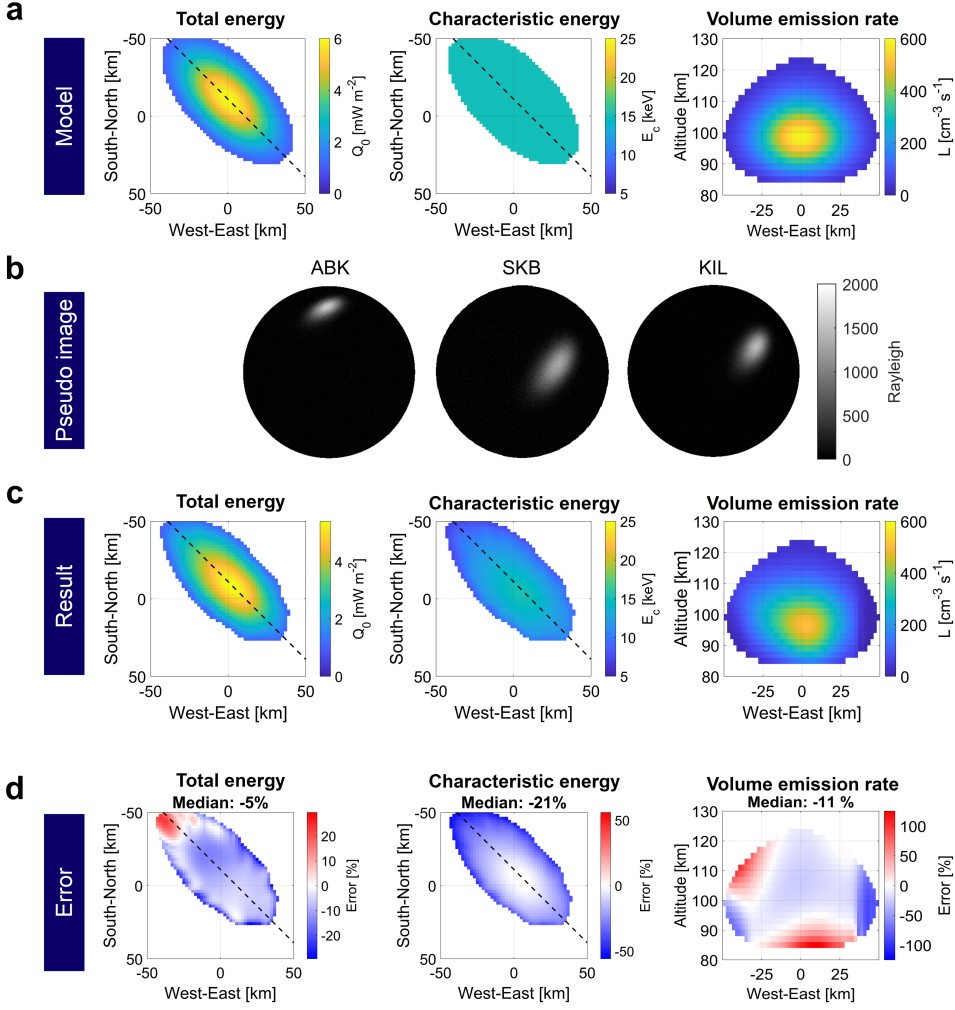

**Figure 2:** (a) The horizontal distribution of the prepared total energy $Q_0$ and the characteristic energy $E_c$ of precipitating electrons and the vertical cross-section of the volume emission rate $L$ along the dashed lines in the left and middle panels. We derived $L$ from the prepared $Q_0$ and $E_c$ values by solving the forward problem. $Q_0$ and $E_c$ are not shown for $Q_0$ values less than $1\,\mathrm{mW\,m^{-2}}$. (b) Pseudo auroral images obtained from model volume emission rates by solving the forward problem. (c) The horizontal distribution of $Q_0$ and $E_c$ and the vertical cross-section of $L$ reconstructed by aurora computed tomography from the pseudo auroral images. (d) The errors of $Q_0$, $E_c$, and $L$, calculated as (Error) = [(Result) − (Model)] / (Model) × 100.





## 3.2 Precipitating electrons

Figures 3a and 3b show $Q_0$ and $E_c$ as reconstructed from the observed auroral images (Fig. 1a). The maximum value of $Q_0$ was ~6 mW m$^{-2}$. The reconstructed $E_c$ ranged from 6 keV to 23 keV. These energies are consistent with observation results from sounding rockets and low-altitude satellites (e.g., McEwen et al., 1981; Miyoshi et al., 2015). We found that the horizontal distribution of $E_c$ was neither uniform nor stable in the patch during the pulsation. In particular, the southwestern part of $E_c$ was enhanced at 00:53:38 UT. It should be noted that the edge and northwestern parts of $E_c$ are expected to be underestimated

due to analytical error, as shown in Fig. 2d. These temporal variations indicate changes in the cyclotron resonance energy of whistler-mode chorus waves during the pulsation. The chorus waves scatter electrons into a loss cone near the magnetic equator. The cyclotron resonance energy of chorus waves depends on the background magnetic field, electron density, and wave frequency (e.g., Kennel and Petschek, 1966). The observed temporal variations thus indicate changes in the magnetospheric source region's background magnetic or plasma environment during the pulsation. Thus, the ACT method is

helpful for investigating PsA-associated temporal variations in the horizontal distribution of precipitating electrons without rocket or satellite observations.

## 3.3 Volume emission rate

Figure 4a shows the 3-D distributions of VERs derived from the reconstructed electron flux by solving the forward problem. Cross-sections in the horizontal plane at an altitude of 94 km are shown in Fig. 4b. The peak altitude ranges from 90 to 104

km (Fig. 4c). The error of the peak altitude is shown in Fig. 4d. The high peak altitude at the northwestern part is expected to be overestimated by at most 8% due to analytical error, as shown in Fig. 2d. The full width at half maximum is almost uniform with a median value of ~20 km (Fig. 4e). For the most part, the altitude width is expected to be overestimated by ~2% (Fig. 4f). The reconstructed peak altitude and width are consistent with those determined in previous studies using stereoscopic observations or an incoherent scatter radar (Brown et al., 1976; Jones et al., 2009; Kataoka et al., 2016). (Stenbaek-Nielsen

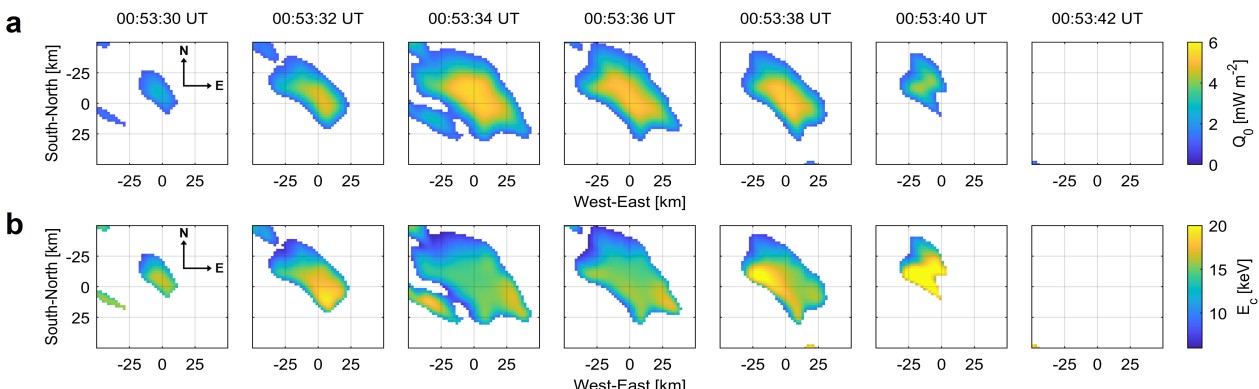

**Figure 3: (a) Total energy, $Q_0$, and (b) characteristic energy, $E_c$, of the precipitating electron flux reconstructed from the observed auroral images. Results of $Q_0$ and $E_c$ where $Q_0$ is less than 1 mW m$^{-2}$ are not shown.**





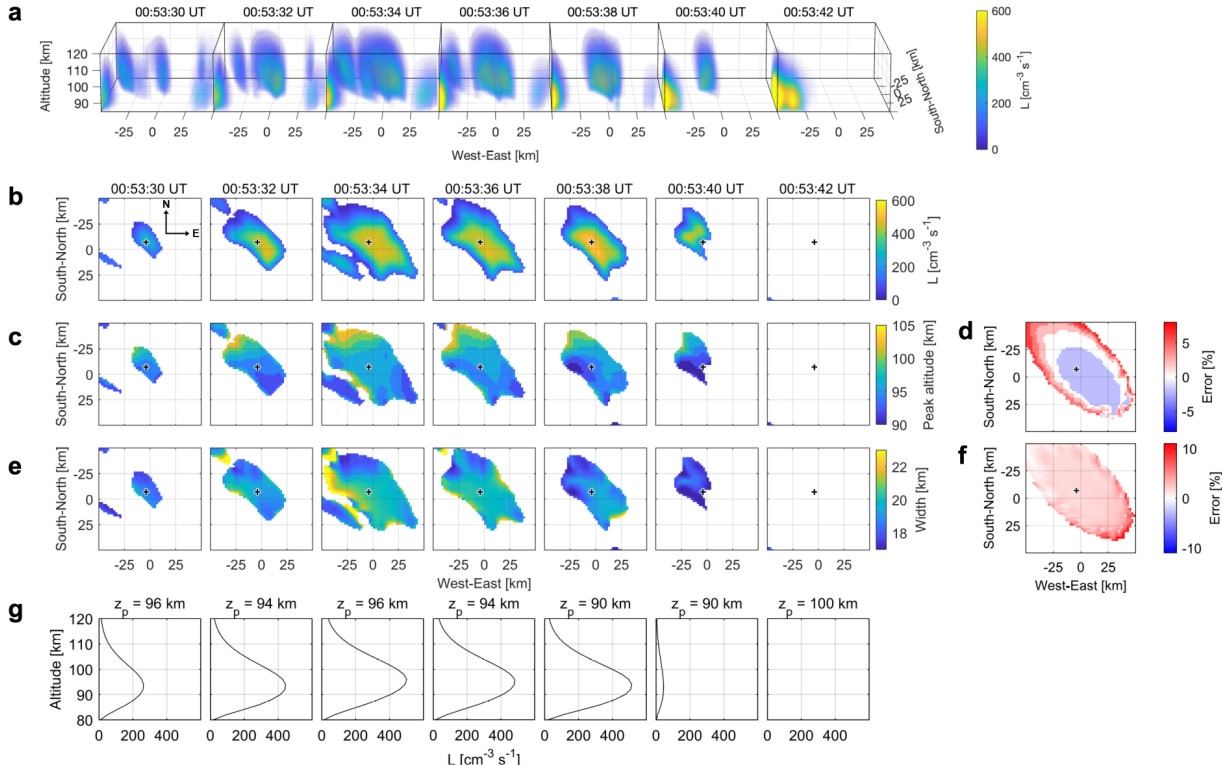

**Figure 4: (a) Reconstructed 3-D distribution of volume emission rates (VERs) $L$. VERs less than 1 cm$^{-3}$ s$^{-1}$ are not shown. (b) Cross-sections in the horizontal plane at an altitude of 94 km. VERs are not shown for $Q_0$ values less than 1 mW m$^{-2}$. (c) Peak altitudes of the reconstructed $L$ and (d) their errors determined using the model aurora. (e) Altitude widths of the reconstructed $L$ and (f) their errors determined using the model aurora. (g) Altitude profiles of $L$ at the European incoherent scatter radar observation point as indicated by black plus marks in Figs. 4b–4f.**

and Hallinan, 1979) reported the existence of thin (<1 km vertical extent) PsA patches based on stereoscopic observations, but our results do not support their results.

The peak altitude of the PsA patch was also estimated by a different. We projected the observed auroral images at altitudes ranging from 80 km to 120 km with an interval of 2 km (Movie S1). The emission altitude was determined to be the altitude at which the residual squared sum between the two projected images reached a minimum value (Fig. S1). The estimated peak

altitude range was 92 to 106 km from 00:53:30 UT to 00:53:40 UT (Fig. S2). These altitudes closely match those determined by ACT.

### 3.4 Electron density

The altitude profiles of VER at the EISCAT radar observation point shown in Fig. 4g were converted to an ionospheric electron density and compared with the actual data observed by the EISCAT radar. The continuity equation for the electron density can

be written as





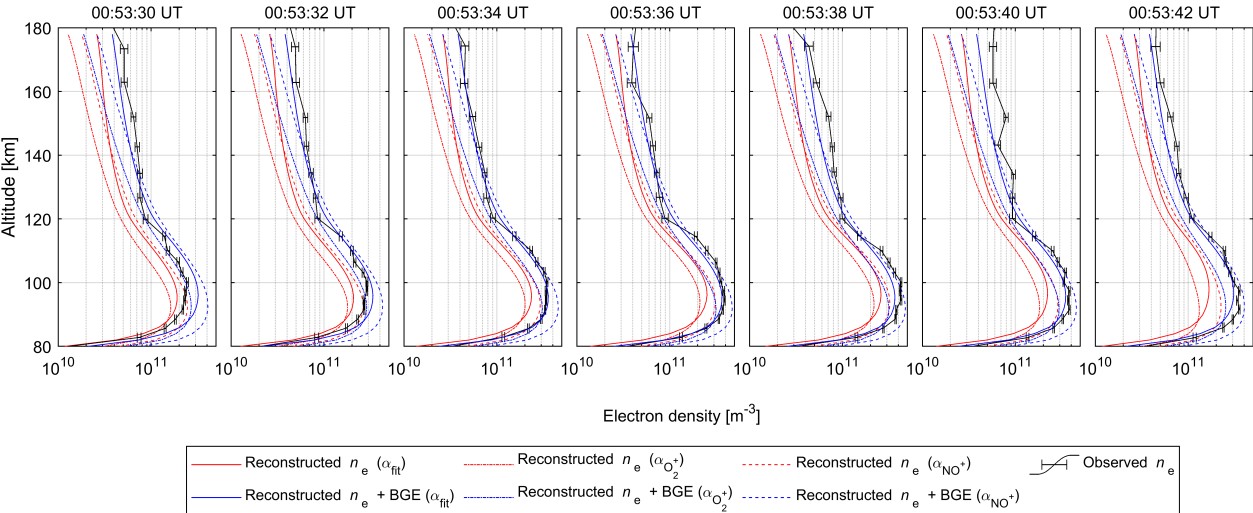

**Figure 5: Electron density ($n_e$) altitude profiles converted from the reconstrucred volume emission rates with the subtraction of the background emission (BGE) (red lines), those without the BGE (blue lines), and those observed by the European incoherent scatter radar (black lines). Details of effective recombination coefficients $\alpha_{\text{fit}}$, $\alpha_{O_2^+}$, and $\alpha_{NO^+}$ are explained in the text. The measurement uncertainties are represented by error bars.**

$$\frac{\partial n_e}{\partial t} = kL - \alpha_{\text{eff}} n_e^2, \tag{4}$$

where $n_e$ [m$^{-3}$] is the electron density, $L$ [m$^{-3}$ s$^{-1}$] is the VER, $k$ is a positive constant for converting VER to the ionization rate (see Appendix), and $\alpha_{\text{eff}}$ [m$^3$ s$^{-1}$] is the effective recombination rate. We derived the electron density from the VER by solving Eq. (4) with the Runge–Kutta method. The initial value was derived from Eq. (4) under steady-state conditions (i.e., $\partial n_e / \partial t =$

0) using reconstructed $L$ at 00:53:36 UT. The VERs were interpolated linearly in order to use the Runge–Kutta method. The altitude profile of $\alpha_{\text{eff}}$ has been investigated by several studies using rocket- and ground-based measurements. Vickrey et al. (1982) summarized many of these results and proposed the following best fit parameterization:

$$\alpha_{\text{fit}} = 2.5 \times 10^{-12} \exp(-z/51.2) \ [\text{m}^3 \text{ s}^{-1}], \tag{5}$$

where $z$ [km] is the altitude. Semeter & Kamalabadi (2005) used the effective recombination coefficients $\alpha_{NO^+}$ and $\alpha_{O_2^+}$ for

NO$^+$ and O$_2^+$, respectively (Walls and Dunn, 1974), as upper and lower bounds on $\alpha_{\text{eff}}$:

$$\alpha_{NO^+} = 4.2 \times 10^{-13} (300/T_n)^{0.85} \ [\text{m}^3 \text{ s}^{-1}], \tag{6}$$

$$\alpha_{O_2^+} = 1.95 \times 10^{-13} (300/T_n)^{0.7} \ [\text{m}^3 \text{ s}^{-1}], \tag{7}$$

Here $T_n$ [K] is the neutral temperature. The red lines in Fig. 5 show the derived electron densities using these three recombination coefficients. We note that these values are underestimated compared to the electron densities observed by the

EISCAT radar (black lines in Fig. 5). This underestimation probably comes from the background emission subtraction from the auroral images prior to ACT and from ambiguity in the effective recombination coefficients. The electron densities reconstructed from auroral images without background emission subtraction are shown as blue lines for reference in Fig. 5.



The reconstruction results from the images, including background emission, approached the electron density profile observed with the EISCAT radar. We noted that the electron density was reconstructed correctly to some extent after the auroral emission
intensity decreased at 00:53:40 UT. This correct reconstruction is due to incorporation of the time derivative term in the continuity equation. The electron density would seem to have rapidly decreased after 00:53:40 UT if the time derivative term were not considered. This result suggests that the time derivative term should be considered when using the continuity equation (Eq. (4)) to derive electron densities associated with PsAs.

It should be noted that the electron density is still underestimated at higher altitudes (>~140 km) even if the background
emission was included. This underestimation would be improved by reconstructing low-energy electron flux from auroral images of various wavelengths (e.g., 844.6 nm).

## 4 Conclusions

We applied the ACT method to PsA patches for the first time and reconstructed the horizontal distribution of precipitating electrons from 427.8-nm auroral images obtained from three observation points. We improved the previously proposed ACT
method by adding the following processes: the subtraction of the background diffuse aurora from the auroral images prior to ACT, the estimation of the relative sensitivity between ASCs, and the determination of the hyperparameters of the regularization term. The characteristic energies of the reconstructed electron fluxes (6 keV to 23 keV) and the peak altitudes of the reconstructed VERs (90 to 104 km) were consistent with those found in previous studies. We determined that the horizontal distribution of the characteristic energy was neither uniform nor stable in the patch during the pulsation, further
underlining the shortcomings of rocket and satellite observations for investigating PsAs. ACT error was evaluated using a model auroral patch. The characteristic energy of electron flux was correctly reconstructed at the center part of the patch but underestimated at the patch edge by at most 56%. The reconstructed electron flux will be improved in future work by incorporating auroral images of various wavelengths.

Although we reconstructed the differential flux of precipitating electrons from auroral images using ACT, Tanaka et al. (2011)
extended ACT to a method called generalized-ACT (G-ACT). G-ACT uses multi-instrument data, such as ionospheric electron density from incoherent scatter radar, cosmic noise absorption from imaging riometers, and the auroral images. They demonstrated that the incorporation of the ionospheric electron density from the EISCAT radar improved the accuracy of the reconstructed electron flux. Furthermore, 3-D ionospheric observation by EISCAT_3D (http://www.eiscat3d.se) is scheduled to begin in 2023. In the future, we will improve the reconstructed electron flux by conducting G-ACT using electron density
data from the EISCAT or EISCAT_3D radar.

## Appendix A

**Estimation of the pulsating auroral emission peak altitude**


Here, we estimate the peak altitude of the PsA patch using a different method from ACT to validate the results from ACT in Sect. 3.3. We projected the observed auroral images at altitudes from 80 km to 120 km with an interval of 2 km. As an example, projected images for 00:53:36 UTC on 18 February 2018 are shown in Movie A1. The emission altitude was determined to be the altitude at which the residual squared sum of the auroral intensity between the two projected images reached a minimum value (Figure A1). The estimated peak altitudes from 00:53:30 UTC to 00:53:40 UTC are shown in Figure A2. These altitudes agreed with the results from ACT in Sect. 3.3.

The link to Movie A1: https://www.dropbox.com/s/gd6jzcr0ghfs7ep/MovieS1_v3.mp4?dl=0

We will issue a DOI for Movie A1 after the acceptance of your manuscript.

**Movie A1: Auroral images projected at altitudes from 80 to 120 km with an interval of 2 km. The images were obtained at Skibotn (SKB), Abisko (ABK), and Kilpisjärvi (KIL) at 00:53:36 UT on 18 February 2018. Residuals between the projected images at each pair of stations (SKB and ABK, and ABK and KIL) are also shown. The auroral intensity is normalized as follows: $I_j = (I - \bar{I})/\sigma$, where $I$ is the auroral intensity, $\bar{I}$ is the average of $I$, $\sigma$ is the standard deviation of $I$, and $j$ signifies ABK, KIL, or SKB. The residual squared sum (RSS) is shown at the top of each panel. The RSS is calculated as $RSS = \sum (I_j - I_{j+1})^2 / N$, where $N$ is the number of datapoints.**

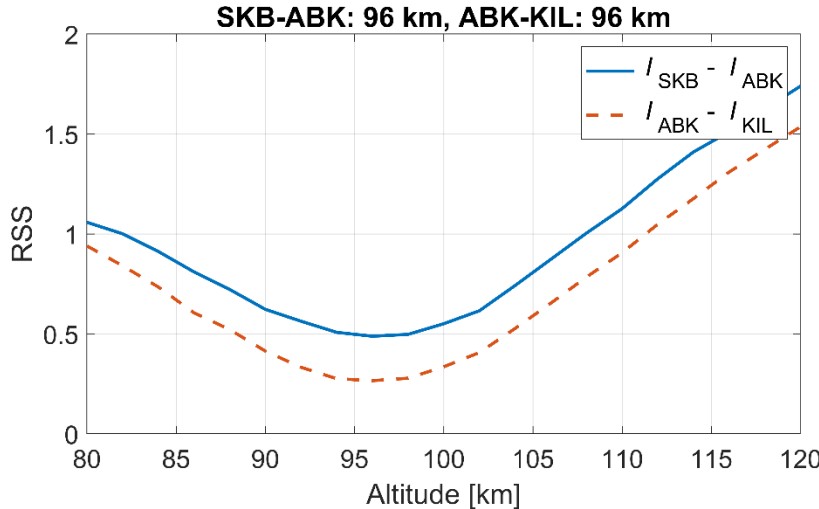

**Figure A1: The residual squared sum (RSS) between the projected images at two stations (SKB and ABK, and ABK and KIL) at each altitude at 00:53:36 UT on 18 February 2018. The altitude at which the RSS reached a minimum value is shown in the panel title.**





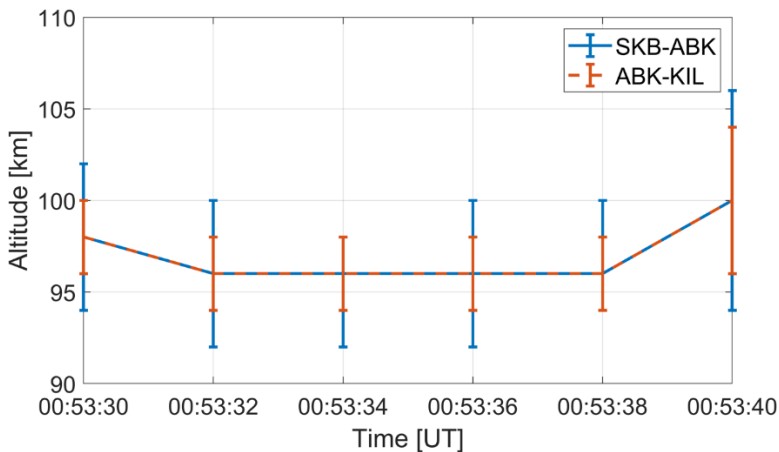

**Figure A2: The altitude at which the residual squared sum (RSS) reached a minimum value for each of 6 timepoints from 00:53:30 UT to 00:53:40 UT on 18 February 2018. Error bars indicate the altitude range over which the RSS was less than 1.2 times each RSS minimum.**

**Appendix B**

**Derivation of $k$**

In this section, we describe how to obtain the positive constant $k(z)$ in Sect. 3.4. The $N_2^+$ (427.8 nm) emission is due to the transition from $N_2^+ \left(B^2\Sigma_u^+\right)_{v=0}$ to $N_2^+ \left(X^2\Sigma_g^+\right)_{v=1}$. According to Sergienko & Ivanov (1993), the VER $L(z)$ [m$^{-3}$ s$^{-1}$] is approximated by

$$L(z) = \frac{A_{0-1}q_{0-0}}{\sum_v A_{0-v}} w(z) = \frac{A_{0-1}q_{0-0}}{\sum_v A_{0-v}} \frac{p(z)\varepsilon(z)}{\Delta\varepsilon}, \tag{B1}$$

where $A_{0-1}$ is the Einstein coefficient for the transition from $N_2^+ \left(B^2\Sigma_u^+\right)_{v=0}$ to $N_2^+ \left(X^2\Sigma_g^+\right)_{v=1}$, $w(z)$ [m$^{-3}$ s$^{-1}$] is the production rate of $N_2^+ \left(B^2\Sigma_u^+\right)$, $q_{0-0}$ is the Franck–Condon factor for the electronic transition from $N_2^+ \left(X^1\Sigma_g^+\right)_{v=0}$ to $N_2^+ \left(B^2\Sigma_u^+\right)_{v=0}$, $p(z)$ is the probability that $\varepsilon(z)$ excites $N_2$, $\varepsilon(z)$ [eV m$^{-3}$ s$^{-1}$] is the energy deposition rate, and $\Delta\varepsilon$ [eV] is the excitation energy cost of $N_2^+ \left(B^2\Sigma_u^+\right)$. The ionization rate due to the precipitating electrons $q_{\mathrm{ion}}(z)$ [m$^{-3}$ s$^{-1}$] is given by

$$q_{\mathrm{ion}}(z) = \frac{\varepsilon(z)}{\Delta\varepsilon_{\mathrm{ion}}} \tag{B2}$$

where $\Delta\varepsilon_{\mathrm{ion}}$ [eV] is the energy used to produce an ion–electron pair. Substituting Eq. (B1) into Eq. (B2) gives

$$q_{\mathrm{ion}}(z) = \frac{\sum_v A_{0-v}}{A_{0-1}q_{0-0}} \frac{\Delta\varepsilon}{\Delta\varepsilon_{\mathrm{ion}}} \frac{1}{p(z)} L(z). \tag{B3}$$

Therefore, the positive constant $k(z)$ for converting VER to the ionization rate is

$$k(z) = \frac{\sum_v A_{0-v}}{A_{0-1}q_{0-0}} \frac{\Delta\varepsilon}{\Delta\varepsilon_{\mathrm{ion}}} \frac{1}{p(z)}. \tag{B4}$$



The parameters used for the calculation are summarized in Table B1.


**Table B1. Simulation parameters used in Eq. (B4).**

| Parameter | Value | References |
|---|---|---|
| $\dfrac{A_{0-1}q_{0-0}}{\sum_v A_{0-v}}$ | 0.197 | A. V. Jones (1974) |
| $\Delta\varepsilon$ | 350 eV | Sergienko & Ivanov (1993) |
| $\Delta\varepsilon_{\text{ion}}$ | 35.5 eV | Semeter & Kamalabadi (2005) |
| $p(z)$ | Calculated from MSISE-00 model | Picone et al. (2002) |

**Data availability**

The EISCAT data are available at http://pc115.seg20.nipr.ac.jp/www/AQVN/evs1.html. The imager data at Skibotn can be obtained at http://pc115.seg20.nipr.ac.jp/www/AQVN/evs1.html.

**Author contribution**

Yoshimasa Tanaka developed the ACT method and code. Yasunobu Ogawa conducted the EISCAT radar observation and prepared the ionospheric electron density data. Kirsti Kauristie, Alexander Kozlovsky, and Tero Raita maintained the

MIRACLE camera network and prepared the auroral images. Mizuki Fukizawa analyzed the data prepared by co-authors and prepared the manuscript with contributions from all co-authors. Takeshi Sakanoi, Keisuke Hosokawa, Björn Gustavsson, Urban Brändström, and Tima Sergienko contributed to the discussion and interpretation of the analysis results.

**Competing interests**

The authors declare that they have no conflict of interest.

**Acknowledgements**

The first author is a Research Fellow (DC) of the Japan Society for the Promotion of Science (JSPS). This study is supported by JSPS KAKENHI Grant Numbers JP17K05672, JP20J11829, and JP21H01152. EISCAT is an international association supported by research organizations in China (CRIRP), Finland (SA), Japan (NIPR), Norway (NFR), Sweden (VR), and the United Kingdom (UKRI). We thank Kellinsalmi Mirjam and Carl-Fredrik Enell for maintaining the MIRACLE camera



network and data flow. The database construction for the imager data at Skibotn and the EISCAT radar data has been supported
by the IUGONET (Inter-university Upper atmosphere Global Observation NETwork) project (http://www.iugonet.org/).

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
