# Peer review of "Reconstruction of precipitating electrons and three-dimensional structure of a pulsating auroral patch from monochromatic auroral images obtained from multiple observation points"

_Annales Geophysicae, 2022_

## Author Response (AR2)

**Response to Referee Comment 1**

We are grateful for the valuable comments and suggestions provided by the Reviewer. We have considered all the comments and suggestions and made appropriate changes to the revised manuscript. The revised part of the manuscript is indicated in red font. Please note that the line number used in "Comment" and "Reply" corresponds to that used in the previously submitted "Preprint" file and the "Track-changes" file submitted this time, respectively.

**Comment:**

Line 138: Can you specify what is meant by "the forward problem"? How exactly did you calculate the VER from the observations in this case?

**Reply:**

We thank the reviewer for this thoughtful comment. As explained in lines 78–79, we calculated the VER from model **f** using the aurora emission model (Eq. (3) in Tanaka et al. (2011)). As suggested by the Reviewer, we revised the above sentence in lines 168–169 of the revised manuscript (see below).

"The 3-D distributions of  VERs derived from the reconstructed electron flux was obtained by solving the forward problem described in Section 2 (see Figure 4a)."

**Reference:**

Tanaka, Y. M., Aso, T., Gustavsson, B., Tanabe, K., Ogawa, Y., Kadokura, A., et al. (2011). Feasibility study on Generalized-Aurora Computed Tomography. *Annales Geophysicae*, *29*(3), 551–562. https://doi.org/10.5194/angeo-29-551-2011

**Technical Corrections:**

**Comment:**

Line 147: By a different what?

**Reply:**

We apologize for the error. As suggested by the Reviewer, we inserted "method" after "a different" in line 179 of the revised manuscript (see below).

"The peak altitude of the PsA patch was also estimated by a different method (Appendix A)."
* * *
**Response to Referee Comment 2**

We sincerely thank the Reviewer for their valuable comments and suggestions. We have considered all the comments and suggestions and made appropriate changes to the revised manuscript. The revised part of the manuscript is indicated in red font. Please note that the line number used in "Comment" and "Reply" corresponds to that used in the previously submitted "Preprint" file and the "Track-changes" file submitted this time, respectively.

This paper describes work done to reconstruct the 3D volume emission rate of a pulsating aurora patch, determine the altitude and vertical thickness of the patch, as well as the horizontal distribution of the characteristic energy and flux of the electron precipitation producing the pulsating aurora. Overall the manuscript is well written and the work is worthy of publication. I have some questions for the authors and suggest some minor corrections below. I also suggest the authors consider what is the most important result from the work. Currently it comes across to me as if the technical process of obtaining the results is the most important aspect of the work, but I wonder if the finding that "the horizontal distribution of $E_c$ was neither uniform nor stable" is at least as important, if not more important. If they authors agree, I suggest they draw more attention to the scientific results (as opposed to the technical process, which is nevertheless impressive) in the abstract and conclusions.

**Reply:**

At 00:53:38 UT, the characteristic energy of precipitating electron flux is large at the southwestern edge of the PsA patch (Figure 3). A possible mechanism that enhanced the characteristic energy at the edge is the field aligned current (FAC). The absence of FAC in the PsA patch has been observed because the PsA patch have no shear motion, which is typically observed in discrete aurora (Davis, 1978). However, several studies have shown that the FAC is associated with PsA patches (Fujii et al., 1985; Gillies et al., 2015; Hosokawa and Ogawa, 2010). If the upward and downward FACs flow at the edge of the PsA path, the potential drop due to the upward FAC can accelerate precipitating electrons and enhance their characteristic energy (Sato et al., 2004). The energy spectra of precipitating electrons observed by rockets and satellites do not show a field-aligned acceleration by the potential drop. However, at altitudes lower than that measured by rockets and satellites, Shepherd and Fälthammar (1980) suggested the existence of a potential drop in the lower E region. We note that the total energy was not enhanced at the edge where the characteristic energy region was enhanced (Figure 3). If the FAC exists, both total and characteristic energy should be increased. Therefore, we cannot suggest the existence of the FAC at the PsA patch's edge for this evet. Multievent analysis is needed to examine the FAC in PsA patches. The 3-D current structure in the PsA patch is beyond the scope of this study. Its reconstruction using ACT and EISCAT_3D radar is planned in the future. As suggested by the Reviewer, we provided the relevant clarifications in lines 155–164 of the revised manuscript (see below).

"The characteristic energy can also be enhanced by the field aligned current (FAC). The absence of FAC in the PsA patch has been observed because the PsA patch have no shear motion and the energy spectra of precipitating electrons observed by rockets and satellites did not show a field-aligned acceleration, which are typically observed in discrete aurora (Davis, 1978). On the other hand, several studies have reported the FAC associated with PsA patches (Fujii et al., 1985; Gillies et al., 2015; Hosokawa and Ogawa, 2010). If the upward and downward FACs flow at the edge of the PsA path, the potential drop due to the upward FAC can accelerate precipitating electrons and enhance their characteristic energy (Sato et al., 2004; Shepherd and Fälthammar, 1980). However, we cannot suggest the existence of the FAC at the PsA patch's edge for this evet because the total energy was not enhanced at the edge (Figure 3). If the FAC exists, both total and characteristic energy should be increased. Multievent analysis is needed to examine the FAC in PsA patches. The 3-D current structure in the PsA patch is beyond the scope of this study. Its reconstruction using ACT and EISCAT_3D radar is planned in the future."

**References:**
Davis, T. N.: Observed characteristics of auroral forms, Space Sci. Rev., 22(1), 77–113, doi:10.1007/BF00215814, 1978.

Fujii, R., Oguti, T. and Yamamoto, T.: Relationships between pulsating auroras and field-aligned electric currents, Mem. Natl. Inst. Polar Res. Spec. issue, 36, 95–1003, 1985.

Gillies, D. M., Knudsen, D., Spanswick, E., Donovan, E., Burchill, J. and Patrick, M.: Swarm observations of field-aligned currents associated with pulsating auroral patches, J. Geophys. Res. A Sp. Phys., 120(11), 9484–9499, doi:10.1002/2015JA021416, 2015.

Hosokawa, K. and Ogawa, Y.: Pedersen current carried by electrons in auroral D-region, Geophys. Res. Lett., 37(18), 1–5, doi:10.1029/2010GL044746, 2010.

Sato, N., Wright, D. M., Carlson, C. W., Ebihara, Y., Sato, M., Saemundsson, T., Milan, S. E. and Lester, M.: Generation region of pulsating aurora obtained simultaneously by the FAST satellite and a Syowa-Iceland conjugate pair of observatories, J. Geophys. Res. Sp. Phys., 109(A10), 1–15, doi:10.1029/2004JA010419, 2004.

Shepherd, G. G. and Fälthammar, C.-G.: Implications of extreme thinness of pulsating auroral structures, J. Geophys. Res. Sp. Phys., 85(A1), 217–218, doi:10.1029/ja085ia01p00217, 1980.

**Minor comments:**

**Comment 1:**

Line 51: What is a "typical" field of view?

**Reply 1:**

The word "typical" was not an appropriate expression. We intended to show the overlap of the the field of view of the three cameras. Therefore, we changed "typical" to "common" in line 53 of the revised manuscript (see below).

"These ASCs have a common  field-of-view as shown in Fig. 1b."

**Comment 2:**

Line 56: Did all four WMI CCD cameras have 427.8 nm filters? What exactly do you mean by compositing auroral images from these cameras?

**Reply 2:**

The Reviewer is correct. All four WMI CCD cameras were equipped with 427.8 nm filters. We averaged the four auroral images from these cameras to improve the signal-to-noise ratio by compositing.

**Comment 3:**

Line 63: I think there's a mistake with the energy range here, are you sure it should be 300 eV to 100 eV? Also, I think it would help readers unfamiliar with the method if you could add a very brief explanation of what the energy axis actually means, before stating its range.

**Reply 3:**

We apologize for the error. The correct energy range was 300 eV to 100 keV. As suggested by the Reviewer, we revised the energy range and included the appropriate explanation in lines 67–69 of the revised manuscript (see below).

"We set the energy ($E$) range to extend from 300 eV to 100 keV. The energy axis contains the information on the auroral emission altitude. This is because the higher the electron energy, the lower the stopping height of the precipitating electrons."

**Comment 4:**

Line 75: Determination of the inverse covariance matrix – you say this is the standard deviation from each auroral image, but the standard deviation over which dimension? Do you mean the standard deviation of the 1024 pixel intensities in the 32x32 region? I think this could be explained more clearly.

**Reply 4:**

We sincerely thank the Reviewer for these thoughtful comments. To determine the standard deviation, we calculated the mean value and standard deviation in the central $5 \times 5$-pixel region for 20 images. Next, we derived a regression line between the mean value and the standard deviation. Finally, we converted the grey level at each pixel to the standard deviation using the equation of the derived regression line. The pixel region was then corrected from $32 \times 32$ to $5 \times 5$ pixels. As suggested by the Reviewer, we included the appropriate explanations in lines 81–84 of the revised manuscript (see below).

"To determine the standard deviation at each pixel in an auroral image, we calculated the mean value and standard deviation in the central $5 \times 5$-pixel regions for 120 images. Next, we derived a regression line between the mean value and the standard deviation. Finally, we converted the grey level at each pixel to the standard deviation using the equation of the derived regression line."

**Comment 5:**

Line 80: $c_{SKB}$ fixed at 1 – should each camera not have a different value, to account for their absolute intensity calibration? Or do you already use calibrated images? Also in the next sentence you say $c_j$ was different for each camera, which contradicts $c_{SKB} = 1$, and in the conclusions say you included the relative sensitivity between ASCs, which also contradicts $c_{SKB} = 1$ I think.

**Reply 5:**

We would like to clarify that calibrated images were used in the manuscript. However, the absolute values obtained from the cameras did not correspond completely. Therefore, we modified the grey levels at KIL and SKB to correspond to those at SKB (i.e., $c_{SKB} = 1$).

**Comment 6:**

Line 81: "during" should be "for".

**Reply 6:**

We sincerely apologize for the error. As suggested by the Reviewer, we made the appropriate correction in lines 89–90 of the revised manuscript (see below).

"The summation was conducted for the first term in Eq. (2) since $c_j$ and $\Sigma_j^{-1}$ were different  for the three ASCs."

**Comment 7:**

Line 90: I think forming the plural of delta this way could be confusing – it's probably better to write "over 5 values of δ(...), which were..." (replacing "δ(...)" with the relevant symbols).

**Reply 7:**

We sincerely thank the Reviewer for pointing this out. As suggested by the Reviewer, we revised "5 $\delta(\lambda,\lambda,c,c)$s" to "5 values of $\delta(\lambda,\lambda,c,c)$" in line 99 of the revised manuscript (see below).

"The cross-validation score $\bar{\delta}(\lambda, \lambda_E, c_{ABK}, c_{KIL})$ was calculated by averaging over  five values of $\delta(\lambda, \lambda_E, c_{ABK}, c_{KIL})$, which were obtained by replacing the test set with one of the training sets in turn."

**Comment 8:**

Line 95: Is it valid to assume a constant background in all voxels? Should there not be a strong altitude dependence, if the background is diffuse auroral emission? Presumably the breakdown of this assumption in the altitude dimension would result in an error in the VER altitude profile of the PsA? Perhaps you could comment on this in the paper, or even better see if the altitude profile of the background could be determined from the "off" pulsating periods (although I understand this may be very challenging).

**Reply 8:**

In our analysis, we assumed that the volume emission rate (VER) was uniform horizontally and vertically (i.e., at all voxels). However, even if the VER was dependent on the altitude, the altitude dependence did not affect our reconstruction results.

The linear integration of the VER in the vertical direction (i.e., the zenith angle $\theta = 0°$) is given by equation R1.

$$\Lambda_0 = \int L(z)\ dz. \tag{R1}$$

Where, $L(z)$ is the altitude-dependent. Assuming that diffuse auroras are uniform horizontally, the linear integration of the VER at the zenith angle can be given by equation (R2)

$$\Lambda(\theta) = \int L(z)\ dz\ /\ \cos\theta = \Lambda_0\ /\ \cos\theta = (constant)\ /\ \cos\theta. \tag{R2}$$

Therefore, the horizontal distribution of the background emission intensity (i.e., diffuse aurora) in the auroral image will depend only on the zenith angle $\theta$. It is not dependent on the altitude distribution of the VER. When we subtracted the background emission image from the observed auroral image, we determined the constant value in equation R2, which fitted the $\Lambda(\theta)$ to the

background emission intensity. Therefore, the reconstruction results were not affected by the altitude distribution of the diffuse aurora.

To clarify these points, we added the appropriate discussion in lines 108–113 of the revised manuscript (see below).

"In this analysis, we assumed that the diffuse aurora showed a uniform VER in all voxels; however, the VER of the diffuse aurora depends generally on the altitude. We note that the altitude dependence of the VER did not affect the analysis result. This is because the horizontal distribution of the background emission intensity (i.e., diffuse aurora) in the auroral image was dependent only on the zenith angle $\theta$ ($\propto \cos\theta$). The distribution is not dependent on the altitude distribution of the VER, if the VER is horizontally uniform."

**Comment 9:**

Sect 3.1: Why is the error in the northwest different to the rest of the edge of the patch? On line 129 you say that part of the reconstruction from the observed images is expected to be underestimated, but does the error depend on the patch location and size? i.e. if you repeated the reconstruction from pseudo images with the patch in a different location, would the horizontal distribution of error change?

**Reply 9:**

The error was dependent on the patch location and size. When we repeated the reconstruction from pseudo images with the path in a different location, a change in the horizontal distribution of error was observed. As shown in Figure 1, the optical observation points (ABK, SKB, and KIL) are located at the southeastern direction from the PsA patch at the EISCAT radar observation point (TRO). We prepared pseudo auroral images with an auroral patch that has a similar location and size with an observed auroral patch. Therefore, the reconstruction results from the observed and pseudo images were expected to show similar errors. To clarify these points, we added appropriate sentences in lines 133–137 of the revised manuscript (see below).

"In this particular event, the optical observation points of ABK, SKB, and KIL were located at the southeastern direction from the PsA patch at the EISCAT radar observation point (TRO) (see Figure 1). Therefore, an accurate reconstruction of the VER peak altitude was more difficult at the northwestern part. This was because of the insufficient information on the northwestern part of the PsA patch due to the bias of the optical observation point distribution."

**Comment 10:**

Fig 4: How were the errors in Figures 4d and 4f determined? These are different to the errors in Figure 2.

**Reply 10:**

We sincerely thank the Reviewer for his/her thoughtful comments. We calculated the error between the model and the reconstruction result of volume emission rate (see Figure 2). We also derived the peak altitudes and the altitude widths of the volume emission rate for the model and the reconstruction result. The errors between them are summarized in Figure 4. To clarify these points, we added an appropriate discussion in lines 171–172 of the revised manuscript (see below).

"The errors of the peak altitude and the altitude width in Figures 4d and 4f, respectively, were derived using the model and reconstructed VER shown in Figure 2."

**Comment 11:**

Line 149: "residual squared sum" should be "sum of the squares of the residuals".

**Reply 11:**

We apologize for the error. As suggested by the Reviewer, we changed "residual squared sum" to "sum of the squares of the residuals" in line 181 of the revised manuscript.

**Comment 12:**

Line 148: I believe Movie S1 should be Movie A1, but if not please can you add a link or reference.

**Reply 12:**

As suggested by the Reviewer, we changed the name "Movie S1" to "Movie A1" in line 180 of the revised manuscript.

**Comment 13:**

Line 158: "see Appendix" should be "see Appendix B".

**Reply 13:**

We apologize for the error. As suggested by the Reviewer, we changed "see Appendix" to "see Appendix B" in line 190 of the revised manuscript.

**Comment 14:**

Line 175: It took me a few moments to understand your point here. I think what you are saying is that the time-dependent continuity equation must be solved to determine the electron density, it cannot be determined instantaneously from the VER (which I agree with). I suggest rewording the last few sentences of this paragraph to make it clearer, probably removing "time derivative term" in favor of some other wording. Also I suggest adding the word "even" on line 174 - "... to some extent even after the auroral emission intensity decreased..."

**Reply 14:**

We sincerely thank the Reviewer for the thoughtful comments. As suggested by the Reviewer, we revised the last few sentences of the pertinent paragraph by removing "time derivative term" and adding the word "even" in lines 206–211 of the revised manuscript (see below).

"We note that the electron density was reconstructed correctly to some extent even after the auroral emission intensity decreased at 00:53:40 UT. This correct reconstruction  considered the time change in the continuity equation. The electron density would seem to have rapidly decreased after 00:53:40 UT, if the time change  was not considered. This result suggested that the time change  should be considered ($dn_e/dt \neq 0$ in Eq. (4)) when using the continuity equation  to derive the electron densities associated with PsAs."

**Comment 15:**

I don't think Appendix A is mentioned in the main text, but probably it should be, I guess in section 3.3.

**Reply 15:**

As suggested by the Reviewer, we made appropriate changes in the sentence in line 179 of the revised manuscript (see below).

"The peak altitude of the PsA patch was also estimated by a different method (Appendix A)."

**Comment 16:**

Please note that the filename for Movie A1 is confusingly Movie S1 – probably they are in fact the same thing?

**Reply 16:**

The Reviewer is correct, the movies were the same. As suggested by the Reviewer, we changed the name "Movie S1" to "Movie A1" in line 180 of the revised manuscript.

**Comment 17:**

Line 206: "residual squared sum" should be "sum of the squares of the residuals".

**Reply 17:**

We sincerely thank the Reviewer for pointing out the error. As suggested, we changed "residual squared sum" to "sum of the squares of the residuals" in line 240 of the revised manuscript.